# Research Progress on the Role of Pyroptosis in Myocardial Ischemia-Reperfusion Injury

**DOI:** 10.3390/cells11203271

**Published:** 2022-10-18

**Authors:** Yang Liu, Jing Zhang, Deju Zhang, Peng Yu, Jun Zhang, Shuchun Yu

**Affiliations:** 1Department of Anesthesiology, The Second Affiliated Hospital of Nanchang University, Nanchang 330000, China; 2Key Laboratory of Anesthesiology of Jiangxi Province, Nanchang 330000, China; 3Food and Nutritional Sciences, School of Biological Sciences, The University of Hong Kong, Pokfulam Road, Hong Kong 999077, China; 4Department of Endocrinology and Metabolism, The Second Affiliated Hospital of Nanchang University, Nanchang 330000, China

**Keywords:** myocardial ischemia-reperfusion injury, pyroptosis, nod-like receptor protein 3

## Abstract

Myocardial ischemia-reperfusion injury (MIRI) results in the aggravation of myocardial injury caused by rapid recanalization of the ischemic myocardium. In the past few years, there is a growing interest in investigating the complex pathophysiological mechanism of MIRI for the identification of effective targets and drugs to alleviate MIRI. Currently, pyroptosis, a type of inflammatory programmed death, has received greater attention. It is involved in the MIRI development in combination with other mechanisms of MIRI, such as oxidative stress, calcium overload, necroptosis, and apoptosis, thereby forming an intertwined association between different pathways that affect MIRI by regulating common pathway molecules. This review describes the pyroptosis mechanism in MIRI and its relationship with other mechanisms, and also highlights non-coding RNAs and non-cardiomyocytes as regulators of cardiomyocyte pyroptosis by mediating associated pathways or proteins to participate in the initiation and development of MIRI. The research progress on novel small molecule drugs, clinical drugs, traditional Chinese medicine, etc. for regulating pyroptosis can play a crucial role in effective MIRI alleviation. When compared to research on other mature mechanisms, the research studies on pyroptosis in MIRI are inadequate. Although many related protective drugs have been identified, these drugs generally lack clinical applications. It is necessary to further explore and verify these drugs to expand their applications in clinical setting. Early inhibition of MIRI by targeted regulation of pyroptosis is a key concern that needs to be addressed in future studies.

## 1. Introduction

Acute myocardial infarction (AMI) is a severe and common medical emergency, resulting in irreversible damage to the heart [1]. AMI leads to ischemia and necrosis of the corresponding myocardial area due to acute and persistent stenosis of the underlying coronary arteries [2]. Most of the AMI-affected patients have a history of coronary heart disease, causing the atherosclerotic plaque to rupture and platelets to aggregate, leading to the formation of a thrombus, which ultimately blocks the coronary lumen. The timely restoration of coronary blood flow and reduction in myocardial infarction size are the primary principles of treatment. Reperfusion therapy is a preferred treatment for acute ST-elevation myocardial infarction. Reperfusion therapy with either direct coronary intervention or thrombolytic therapy has already become a very common and mature clinical therapy. However, in the past few years, it has been observed that myocardial reperfusion may lead to more myocardial cell death, thus further deteriorating the cardiac function of the patients and leading to poor treatment effects. This phenomenon is called myocardial ischemia-reperfusion injury (MIRI). The four different clinical manifestations of MIRI include myocardial stunning, no-reflow phenomenon, reperfusion arrhythmia, and lethal reperfusion injury [3,4]. Currently, it is accepted that MIRI not only includes necrosis, but also involves several pathophysiological changes, such as apoptosis, autophagy, endoplasmic reticulum stress, ferroptosis, pyroptosis, and other mechanisms that interact and interfere with cell signaling, ultimately leading to intracellular calcium overload, aggravation of oxidative stress, and activation of inflammatory cascades [5,6,7,8].

Pyroptosis, a form of lytic programmed cell death induced by inflammasomes, has gradually attracted more attention in recent years. Previous studies have shown that it was involved in the pathogenesis of many cardiovascular diseases and also played a vital regulatory role in the occurrence and development of MIRI. Therefore, elucidating the pyroptosis mechanism could help in MIRI alleviation by targeting this pathway [9,10].

## 2. Mechanism of Pyroptosis in MIRI

Since its discovery, pytoptosis has been thoroughly studied for a long time. Zychlinsky et al. discovered a Shigella flexneri-mediated cytotoxic killing mechanism in 1992 after human colon mucus was invaded by the Gram-negative bacterial pathogen. At that point in time, this was regarded as an invasive strain-induced programmed cell death and was characterized by chromatin condensation, DNA fragmentation, and a caspase-dependent apoptosis pathway [11].

In 2001, “pyroptosis” was defined as caspase-1-dependent pro-inflammatory programmed cell death [12]. The 2018 Nomenclature Committee on Cell Death (NCCD) regarded pyroptosis as a regulated cell death primarily depending on the pores in the plasma membrane formed by the members of the gasdermin protein family, which are generally formed by caspase activation [13].

Pyroptosis is the mechanism by which the human immune system responds to pathogens as well as endogenous damage. Nod-like receptor protein 3 (NLRP3) senses cardiomyocyte injury and recruits an apoptosis-associated speck-like protein containing a carboxy-terminal CARD (ASC) and procaspase-1 to form inflammasomes, which initiate the pyroptosis pathway. This, in turn, activates gasdermin D (GSDMD), which triggers cell death, and is also the classical pathway of pyroptosis [14].

The NLRP3 inflammasome is an important component of innate immune response-dependent pattern-recognition receptors and is a key protein that triggers pyroptosis in MIRI. Toll-like protein-sensing receptors derived from danger-associated molecular patterns (DAMPs) activate the nuclear factor kappa B (NF-κB) pathway, K^+^ efflux, and lysosome destruction, while the changes in both calcium concentration and reactive oxygen species (ROS) production can lead to the activation of the NLRP3 to form NLRP3 inflammasomes. Moreover, upon activation, the NLRP3 protein recruits ASC and cysteine-recruiting proteins to complete the inflammasome assembly, and subsequently activates caspase-1 [15,16,17]. On the one hand, activated caspase-1 cleaves a GSDMD to generate an N-terminal amino peptide and C-terminal carboxyl peptide [18]. The N terminus is the lipophilic region that interacts with the lipid layer on the inner side of the cell membrane, resulting in the formation of pores in the cell membrane, increase in the cell permeability, and release of cellular contents, thereby resulting in an inflammatory response. Meanwhile, the changes in cell permeability generally induce the entry of extracellular fluid into the cell, causing cell lysis. The C-terminus is hydrophilic and can be used as an inhibitory region [19,20,21]. On the other hand, the precursors of interleukin (IL)-1β and IL-18 are cleaved by activated caspase-1 to active mature IL-1β and IL-18, respectively. These two precursors are released extracellularly upon cleavage and could recruit inflammatory cells to further trigger the inflammatory response [20].

## 3. Relationship between Pyroptosis and Other Associated Mechanisms of MIRI

### 3.1. Oxidative Stress

During the process of myocardial ischemia and reperfusion, a large amount of ROS is produced, especially in the initial stage of reperfusion. The excessive ROS induces cell damage by disrupting the cellular signaling pathway, activating the inflammatory factors, and inducing lipid peroxidation and even cell death [22]. The massive accumulation of ROS stimulates cardiomyocytes by multiple mechanisms, thereby leading to cardiomyocyte necrosis or activation of redox signaling pathways to induce cardiomyocyte apoptosis [23,24]. The ROS generation in mitochondria are specific drivers regulating the damage mechanisms in MIRI and can induce a mitochondrial permeability transition, resulting in and then cause oxidative damage to mitochondrial structures and molecules [25]. Recent studies have shown that, during the process of myocardial ischemia and reperfusion, ROS can stimulate tissue inflammation and NLRP3 inflammasome activation in the brain, heart, kidney, and testis. ROS production may be directly induced by NLRP3 in functional or dysfunctional mitochondria or indirectly affected by other NLRP3 activators [26]. ROS can also stimulate the production of IL-18 in MIRI, thereby leading to inflammation in tissues and increasing apoptosis and calcium overload, ultimately resulting in myocardial injury [27,28]. ROS activates certain cell factors in the pyroptosis pathway to aggravate cell damage. Moreover, pyroptosis also promotes ROS production by releasing several inflammatory factors.

### 3.2. Calcium Overload

During MIRI, dysfunctional calcium homeostasis and disordered calcium distribution are observed in the cells, resulting in an abnormal increase in intracellular calcium concentration, also known as calcium overload. Calcium overload can disturb of the oxidative phosphorylation cycle in mitochondria, thereby leading to a decrease in mitochondrial membrane potential, a decrease in tissue ATP content, and metabolic disturbance in cardiomyocytes.

Firstly, calcium overload is shown to be closely related to pyroptosis in other non-cardiomyocytes. It was observed that the accumulation of inflammatory metabolites in chondrocytes of joint tissues in rats led to a decrease in the PH of the joint tissues and activation of acid-sensing ion channel 1a, which promoted the influx of calcium ions into cells, thereby leading to increased intracellular calcium ion concentrations. This, in turn, stimulated NLRP3 expression, polymerization, and inflammasome assembly. These changes further lead to the activation of caspase-1 by cleavage of pro-IL-1β and pro-IL-18 into their biologically active forms, followed by the initiation of pyroptosis. Therefore, it can be concluded that the increased intracellular calcium ion concentration promoted cell pyroptosis by upregulating the expression of the inflammasome [29]. In addition, Zhou et al. reported that local anesthetics activated calcium/calmodulin-dependent protein kinase II via the phosphorylation of the ion channel TPRV1, and also further led to calcium overload, reduction in the mitochondrial membrane potential, and an increase in the release of caspase-3. This, in turn, cleaves gasdermin E, thereby initiating the pyroptosis of glioblastoma cells. Further, this study also demonstrated that the pyroptotic pathway can be initiated by the calcium overload pathway [30].

Significant progress has been gradually made to understand the protective effect of calcium overload-mediated pyroptosis in cardiomyocytes.

Mo et al. developed an adult rat cardiomyocyte hypoxia/reperfusion (H/R) model and observed that calcium overload could result in H/R-induced cardiomyocyte pyroptosis by regulating the NLRP3/caspase-1 pathway. A study observed that the intracellular ion channel inositol 1,4,5-triphosphate receptor IP3R1 activated calcium overload by increasing the intracellular calcium ion concentration and also aggravated pyroptosis caused by MIRI [31]. Aconitine-induced calcium overload can be alleviated by Ginsenoside Rb1 via effectively restoring calcium homeostasis and reducing myocardial cell damage by inhibiting cell pyroptosis and calcium transients in rat ventricular myocytes [32].

Particularly, calcium overload in MIRI acts as an excitatory factor of oxidative stress, which further stimulates the formation of inflammasomes. Moreover, it has been proven that oxidative stress and pyroptosis promote each other during MIRI. Thus, it is evident that there is a strong relationship between them during the progression of MIRI.

### 3.3. Apoptosis and Necroptosis

So far, the relationship between pyroptosis, apoptosis, and necroptosis in the MIRI model has not yet been fully explored. However, current studies have indicated that the cell death-regulated pathways can cross-regulate each other [33], and caspase-8 acts as an important molecular switch that controls them [34,35,36,37].

Caspase-8 is a protease that regulates both cell death and survival. The extrinsic pathway of apoptosis is triggered by the transmembrane death receptors belonging to the tumor necrosis factor (TNF) family. Caspase-8 is involved in executing this extrinsic pathway [14]. Moreover, caspase-8 was shown to inhibit RIPK3/RIPK1 and MLKL-mediated necroptosis [38,39], with the former considered to be a typical signaling module of necroptosis [40].

Recently, the role of caspase-8 in pyroptosis was demonstrated in embryonic intestinal tissue. Briefly, the deficiency of caspase-8 induced ASC formation, thereby upregulating GSDMD, caspase-3, and caspase-1-dependent cleavage of caspase-7. In other words, higher levels of caspase-8 could suppress pyroptosis-related proteins by inhibiting ASC formation [41]. Caspase-8 is a bridge linking pyroptosis with apoptosis and necroptosis. Fas-associated death domain (FADD) and caspase-8 can act as apical mediators in regulating the NLRP3 inflammasome activation, and subsequently, affect the expressions of caspase-1 and IL-1β downstream of NLRP3 [42]. Caspase-8 can be regulated by the apoptosis upstream of FADD and is involved in the pyroptotic inflammasome activation, which is an extrinsic apoptosis pathway. Additionally, caspase-8 also plays a critical role in the intrinsic (mitochondrial) pathway, which is the other apoptosis pathway primarily regulated by the B-cell lymphoma 2 (Bcl-2) family proteins in the outer mitochondrial membrane. The Bcl-2 family proteins determine the permeability of the mitochondrial membrane and also play an important role in regulating the release of Cytc. In terms of functionality, these proteins can be divided into pro-apoptotic and anti-apoptotic. Vince et al. demonstrated that the mitochondrial apoptotic effectors BAX/BAK activated caspases-3 and -7 to trigger NLRP3 inflammasome and caspase-8-driven IL-1β activation [43]. The same key molecule of necroptosis, RIPK3, can also lead to the rapid and complete production of active IL-1β by NLRP3-caspase-1 inflammasome or caspase-8 [44,45]. It can be concluded that caspase-8 links pyroptosis, apoptosis, and necroptosis, and is regulated by both apoptosis and necroptosis, thereby affecting pyroptosis.

The above findings demonstrated that apoptosis, necroptosis, and pyroptosis are overlapping pathways and exhibit overlapping effects in coordinating programmed cell death. That indicates that, if one pathway is activated or deactivated, other pathways would also be affected due to a chain reaction (Figure 1). Therefore, the research focus should be on the combination of common action sites of the multiple pathways and understanding the mechanism of the switch regulator caspase-8. Further studies should attempt to identify more signaling pathway molecules that affect the three pathways at the same time and have a protective effect on MIRI.

There are few studies on the relationship between pyroptosis and other mechanisms of MIRI, such as autophagy, endoplasmic reticulum stress, and ferroptosis. Thus, those studies are not considered here.

## 4. Effect of Pyroptosis on Non-Cardiomyocytes and Its Role in MIRI

Pyroptosis has been shown to act on non-cardiomyocytes, thereby leading to MIRI. Pyroptosis aggravates MIRI by acting on non-myocardial cells, including fibroblasts, vascular endothelial cells, and macrophages, to activate the inflammatory pathways.

### 4.1. Fibroblasts

Fibroblasts are the central cells producing extracellular matrix and account for 60% to 70% of the total cells in the normal myocardial tissue. They are also the major constituent of non-cardiomyocytes in the heart. A study by Kawaguchi demonstrated that the activation of inflammasomes by cardiac fibroblasts is critical for MIRI. Importantly, in vitro experiments revealed that H/R stimulated inflammasome activation in cardiac fibroblasts and not in cardiomyocytes. This activation was mediated by the production of ROS and potassium efflux, thereby inducing the secretion of IL-1β, which ultimately led to cardiac fibroblast pyroptosis. Cardiac fibroblasts are more capable of detecting injury signals than cardiomyocytes, and thus enhance the inflammatory response to MIRI [46]. In addition, the NLRP3 inflammasome is upregulated in cardiac fibroblasts and mediates MIRI. After NLRP3 was knocked down in fibroblasts of rats, the contractile function of the hearts was preserved after in vitro ischemia reperfusion, and the infarct size and apoptosis were significantly reduced compared to the control. It is noteworthy that, although both ASC and NLRP3 are parts of pyroptosis inflammasome, the infarct sizes of NLRP3-deficient hearts differ from those of the ASC-deficient hearts, while no significant differences in infarct size were observed between ASC-deficient hearts and wild rat hearts. Therefore, further studies on understanding the mechanisms underlying the differential protective effects of NLRP3 and ASC are necessary [47,48].

### 4.2. Vascular Endothelial Cells

The leukocyte aggregation and activation mediated by microvascular injury play an important role in MIRI. The expressions of various adhesion molecules of neutrophils and vascular endothelial cells are enhanced, and this leads to high adhesion and aggregation of these cells. Neutrophils can chemotactically migrate through the vascular wall, thereby aggravating the infiltration of inflammatory cells. The free radicals and lysosomal enzymes cause damage to the endothelial, eventually forming microthrombi and tissue edema, and even leading to a no-reflow phenomenon [49,50]. The vascular endothelial cells participate in inflammatory responses, and endothelial dysfunction is a marker and directly responsible for several cardiovascular diseases or adverse cardiovascular events. A large amount of ROS released during MIRI can lead to the activation of the NLRP3 inflammasome, thereby aggravating inflammation and cell damage. The role of NLRP3 inflammasome activation in endothelial dysfunction was found to induce pyroptosis. In recent years, a large number of studies have shown that non-coding RNAs (ncRNAs) may regulate endothelial function by mediating the NLRP3 inflammasome signaling pathway. The microRNA-129 overexpression in endothelial cell-derived extracellular vesicles affected the inflammatory response caused by MIRI [51]. The microRNA-495 improved cardiac microvascular endothelial cell injury and inflammation response by suppressing the NLRP3 inflammasome signaling pathway, thereby alleviating MIRI [52,53]. Similarly, various methods have been used for the protection of endothelial cells and attenuation of their oxidative stress by suppressing the activation of the NLRP3 inflammasome. Adiponectin alleviates NLRP3-mediated pyroptosis of endothelial cells by inhibiting FoxO4 in human aortic epithelial cells [54]. It is expected that more drugs can alleviate MIRI by inhibiting the pyroptosis of vascular endothelial cells in the future. Thus, the discovery and development of such drugs would be worthy of further investigation.

### 4.3. Macrophages

MIRI leads to a release of large amounts of cardiac extracellular vesicles (EVs), which, in turn, promote local aseptic inflammation and can be shunted to distant organs. EVs can deliver miR-155-5p to M1 macrophages, promote M1-like macrophage polarization, increase the expressions of pro-inflammatory cytokines, and induce a pro-inflammatory phenotype by activation of the JAK2/STAT1 pathway [55]. Macrophages are the central mediators of cardiac inflammation. They are not only involved in the initiation and resolution of the inflammatory response, but also tissue repair. Thus, targeting macrophages may be a potential intervention for MIRI. Dai et al. demonstrated that M2 macrophage-derived exosomes carrying microRNA-148a alleviate MIRI by suppressing TXNIP and TLR4/NF-κB/NLRP3 inflammasome signaling pathways [56].

Several studies have demonstrated that exosomes may activate the cardioprotective signaling pathway [57]. For example, Yue et al. observed that mesenchymal stem cell-derived exosomal microRNA-182-5p alleviated MIRI by targeting GSDMD in rats [58]. The aforementioned non-cardiomyocyte-derived exosomes can carry micro RNAs (miRNAs) to improve cell pyroptosis by inhibiting the NLRP3 inflammasome activation pathway. Therefore, at present, more studies on stem cell-derived exosomes in the pyroptotic pathway are necessary, and further investigations on the amelioration of MIRI by ncRNAs via the inflammasome signaling pathway are underway.

## 5. Regulation of Pyroptosis by Non-Coding RNAs in MIRI

In recent years, epigenetics has gradually become a research hotspot, and several studies have revealed that it plays an important role in MIRI regulation and pathogenesis. Moreover, a large of studies have indicated that ncRNAs are involved in MIRI regulation. They can be used as potential biomarkers and therapeutic targets for MIRI. Moreover, they also have broad prospects in disease diagnosis and clinical treatment [59]. NcRNAs consist of long non-coding RNAs (lncRNAs), miRNAs, and circular RNAs (circRNAs).

### 5.1. MicroRNA

The miRNAs are non-coding single-stranded RNA molecules with a length of approximately 22 nucleotides and are involved in the regulation of post-transcriptional gene expression. The miRNAs regulate the expression of different genes by inhibiting the translation of messenger RNA (mRNA), thereby promoting mRNA degradation [60]. The expression of miR-29a was significantly upregulated in cardiomyocytes affected by MIRI, and SIRT1 was demonstrated to be a direct target of miR-29a. The inhibition of miR-29a improved MIRI by targeting SIRT1 and suppressing oxidative stress and NLRP3-mediated pyroptosis [61]. Zhou et al. observed that inhibition of miR-132 could improve MIRI by targeting SIRT1 to activate the PGC-1α/Nrf2 signaling pathway for suppressing oxidative stress and pyroptosis. Moreover, the expressions of related proteins NLRP3, caspase-1, and IL-1β were also found to decrease significantly [62]. The miRNA expression plays an important role in MIRI by regulating pyroptosis as well various other pathways such as apoptosis and necrosis. However, the development of miRNA therapeutics for clinical applications is a challenge, and the development of miRNA mimics and inhibitors still requires a lot of effort in the future.

### 5.2. LncRNA

The lncRNAs are non-coding RNAs having a length of more than 200 nucleotides. The abnormal expression of lncRNAs in the cardiovascular system is associated with abnormal cardiac development and cardiovascular diseases [63,64]. The LncRNAs also induce MIRI through oxidative stress, immunity, and mitochondrial biological functions. Several studies have demonstrated that lncRNAs can regulate the expression of downstream proteins, thereby affecting pyroptosis and mediating MIRI. The different lncRNAs have varying expression levels during MIRI. Therefore, their regulation of pyrotosis by these lncRNAs may differ drastically, and they generally mediate the development and progression of MIRI by regulating the expression of miRNAs. For example, lncRNAs Rian overexpression reduces cardiomyocyte pyroptosis and decreases the expressions of inflammatory molecules in the associated cell pyroptosis pathways. Furthermore, miR-17-5p bound to lncRNAs Rian was demonstrated to promote CCND1 transcription, reduce cardiomyocyte pyroptosis, and alleviate MIRI [65]. In addition, lncRNA ROR was shown to induce pyroptosis and inflammatory responses by inhibiting the miR-185-5p expression in H/R cardiomyocytes. On the other hand, the overexpression of miR-185-5p was shown to reverse H/R-induced cell pyroptosis and upregulation of LDH, IL-1β, and IL-18, while the overexpression of CDK6 negatively regulated miR-185-5p. This indicated that the lncRNA ROR/miR-185-5p/CDK6 axis could regulate MIRI [66]. However, it has been revealed that GSDMD-mediated pyroptosis ameliorated MIRI, inhibiting lncRNA PVT1 expression [67]. Thus, lncRNAs can regulate the related miRNAs and play a pivotal role in MIRI, which can help to provide more therapeutic options for MIRI treatment.

### 5.3. CircRNA

The circRNAs are endogenous biomolecules forming covalently closed loops with tissue- and cell-specific expression patterns in eukaryotes, and their biogenesis is regulated by the specific cis-acting elements and trans-acting factors. Several circRNAs exert important biological effects by acting as miRNAs or protein inhibitors (also known as “sponges”), thereby regulating protein functions or gene expression [68]. Recent studies have shown that circRNAs are rich in miRNA binding sites and act as miRNA sponges in cells, thereby exhibiting inhibitory effects on their target genes, leading to increased target gene expression levels. This mechanism known as the competing endogenous RNA (ceRNA) mechanism. The circRNAs interact with disease-associated diseases. Several studies have demonstrated that circRNAs function in MIRI by combining with miRNAs, and this was still mainly evident in the regulatory effect of circRNAs on apoptosis [69,70,71]. Only a few studies investigate the relationship between circRNAs and pyroptosis. One study by Ye et al. investigated the role of circ-NNT/miR-33a-5p/USP46 signaling axis in mediating MIRI and observed that circ-NNT regulates USP46 by sponging miR-33a-5p, thereby promoting pyroptosis and MIRI. The binding sites with a “sponge effect” may serve as therapeutic targets for mitigating MIRI [72].

Therefore, several nc RNAs regulate pyroptosis in MIRI. (Table 1). In addition, previous studies have demonstrated that the three nc RNAs were regulated by axial feedback, and the investigation of the signaling pathways can contribute to the clinical development of targeted therapy for MIRI.

## 6. Inhibition of Pyroptosis by Drugs and Improved MIRI

### 6.1. Small Molecular Substances

MCC950 is a small molecule inhibitor that selectively blocks NLRP3 inflammasome activation [75]. Moreover, the targets of MCC950 may be associated with chloride efflux, CLICs, or other targets acting upstream of chloride efflux. Several studies have reported the reduction in NLRP3 inflammasome activation by MCC950 inhibition of chloride efflux-dependent ASC [76]. In a porcine myocardial ischemia-reperfusion model, depending on the dose, MCC950 reduced myocardial infarct size and preserved myocardial function. However, no significant differences were observed in cardiac function before and after reperfusion. This indicated that MCC950 may lead to reduction in the inflammatory response in the subacute phase after myocardial infarction. A study indicated that MCC950 reduced circulating markers of injury and inflammation, and also exhibited a lower level of neutrophil infiltration in the myocardium. However, a higher level of monocyte infiltration was obtained in MCC950-treated animals, which may be caused by increased infiltration of anti-inflammatory macrophages [77].

INF4E is also a potent NLRP3 inflammasome inhibitor and can inhibit the activities of caspase-1 and NLRP3-ATPase. A study by Mastrocola et al. demonstrated that the time-dependent attenuation of MIRI by INF4E via activation of reperfusion injury rescue kinase (RISK) and mitochondrial pathways [78]. Further, they observed that INF4E preconditioning reduced myocardial infarct size, improved recovery of myocardial contractility, inhibited NLRP3 inflammasome and downstream signaling pathway, enhanced activity of RISK protective pathway, and improved mitochondrial biogenesis and energy metabolism. INF4E was also found to significantly reduce GSDMCD1 cleavage and IL-1β release. Both the ischemia-reperfusion injury and drug treatment had no significant effects on IL-1β mRNA levels, thus highlighting the selective influence of INF4E on the NLRP3 inflammasome-dependent IL-1β cleavage, and there was no effect on the IL-1β expression. However, that study could not elucidate whether INF4E acts on cardiac fibroblasts, particularly cardiomyocytes to a limited degree. Recent studies have reported an inhibitor of NLRP3 more specific than INF4E, i.e., INF39, and the results indicated that INF39 attenuated NLRP3 assembly in macrophages. However, the detailed mechanism of INF39 and direct targets of its anti-inflammatory activity have yet to be studied. This can help further evaluate its applications in myocardial ischemia-reperfusion models [79]. The focus of future research must be on understanding the mechanism of action of drugs in specific cardiomyocytes.

Generally, VX-756 is metabolized to the active molecule VRT-043198, a highly selective prodrug caspase 1 inhibitor and a potent caspase-1 selective inhibitor.

During reperfusion, the co-administration of VX-765 with ticagrelor or cangrelor (P2Y12 receptor antagonists) led to a reduction in the myocardial area and an improvement in ventricular function in an ischemia-reperfusion mouse model. VRT-043198, an active derivative of VX-765, improved myocardial function at the onset of reperfusion, implying that caspase-1 tends to damage the heart only upon reperfusion. Furthermore, the studies observed that VX-765 reduced circulating IL-1β, prevented the loss of cardiac glycolytic enzymes, preserved mitochondrial complex I activity, and suppressed the release of lactate dehydrogenase release [80,81]. Carmo et al. observed that VX-765 had a protective effect on the isolated rat hearts through the RISK pathway. However, they did not observe any additional myocardial protection in combination with ischemic preconditioning [82]. The RISK pathway refers to a group of pro-survival protein kinases, which confer cardioprotection when activated specifically during reperfusion. This pathway could utilize growth factors to aggravate the apoptotic process by activation of pro-survival proteins, such as PI3K-Akt and MEK1-ERK1/2 [83]. Thus, it can be concluded that a close relationship between pyroptosis and apoptosis exists, and the small molecules exhibit a protective effect on MIRI by inhibiting these pathways. It is expected that based on this mechanism, certain small molecule drugs can be developed to treat and prevent MIRI and reduce the cardiomyocytes damage in MIRI.

### 6.2. Clinical Drugs

Colchicine leads to a reduction in the activity, adhesion, and chemotaxis of neutrophils by interfering with the lysosomal degranulation, and it also inhibits the granulocyte migration to the inflammatory area, thereby exerting an anti-inflammatory effect. In the past few years, colchicine has been demonstrated to exhibit anti-inflammatory effects and has been proved to be effective in a wide range of cardiovascular diseases [84]. Several clinical and basic experiments have confirmed that colchicine plays an important role in MIRI prevention. Randomized controlled studies have indicated that colchicine at a dose of 0.5 mg per day led to a significantly lower risk of ischemic cardiovascular events compared to placebo and reduced the costs of standard care after myocardial infarction [85,86].

In this study, the focus is mainly on the anti-inflammatory and anti-pyroptotic effects of colchicine in the MIRI model. Colchicine was found to be associated with increased levels of systemic interleukin-10 (IL-10) and decreased levels of cardiac transforming growth factor-β [87]. The pretreatment with colchicine reduces myocardial infarct size and induces cardioprotection associated with anti-inflammatory effects during the early reperfusion phase. In fact, colchicine can lead to MIRI inhibition through multiple mechanisms and MIRI-induced myocardial injury inhibition by a reduction in caspase-3 levels and activation of the PI3K/AKT/eNOS pathway in H9C2 cells undergoing apoptosis in a hypoxia/deoxygenation-induced myocardial injury model [88]. Moreover, it also plays important role in sympathetic denervation [89] and reduction in cardiac fibrosis [90].

Metformin is the first-line drug for the treatment of type 2 diabetes, and several studies have demonstrated that it plays a crucial role in ischemia-reperfusion injury in different organs such as the heart, gut, brain, kidney, bladder, testis, ovary, and liver [91,92,93,94,95,96,97]. Generally, metformin inhibits apoptosis and autophagy and reduces oxidative stress, mainly by regulating the signal transduction pathways [98,99]. Recent clinical studies have indicated that metformin improves chronic inflammation by the improvement metabolic indicators, such as hyperglycemia, insulin resistance, and atherosclerotic dyslipidemia, and also has direct anti-inflammatory effects on MIRI. In particular, metformin has been demonstrated to suppress the inflammation mainly by inhibition of NFκB through AMP-activated protein kinase (AMPK)-dependent and -independent pathways [100,101,102]. A study by Zhang et al. demonstrated that metformin regulated the inflammatory response induced in MIRI to exert cardioprotective effects by enhancing the AMPK pathway and inhibiting NLRP3 inflammasome activation [103]. The Langendorff rat heart model with MIRI and neonatal rat ventricular myocyte model were established to mimic the isolated heart and in vitro cellular environment. The results from this study indicated that the post-treatment with metformin can significantly lead to a reduction in infarct size, alleviation of apoptosis, and inhibition of myocardial fibrosis. Furthermore, metformin was found to activate phosphorylated AMPK; decrease proinflammatory cytokines TNF-α, IL-6, and IL-1β; and also lead to a further decrease in the NLRP3 inflammasome activation.

Dexmedetomidine is a high selective α2-adrenoceptor agonist that is commonly used for perioperative sedation. Dexmedetomidine can reduce MIRI and exerts a protective effect on ischemia-reperfusion injury by improving myocardium function, which is not regulated by the central nervous system [104,105]. A study by Zhong et al. found that the enhanced expression of NLRP3, ASC, and cleaved caspase-1 triggered pyroptosis, whereas miR-29b enhanced both ischemia-reperfusion and H/R-induced cardiomyocyte pyroptosis in the ischemia-reperfusion rat model. FoxO3a acts as a target for miR-29b and inhibits cardiomyocyte pyroptosis by regulating ARC. Dexmedetomidine can alleviate MIRI and H/R injury in rats and ameliorate the pyroptosis in myocardial ischemia-reperfusion rats and H/R-injured cardiomyocytes by downregulating miR-29b for the activation of the FoxO3a/ARC axis [73].

Trimetazidine is an inhibitor of free fatty acid oxidation playing a crucial role in myocardial and muscle glucose metabolism and has been demonstrated to provide cardioprotection without any associated side effect in patients with angina, diabetes, and undergoing revascularization surgery [106,107]. Trimetazidine alleviates MIRI-induced pyroptosis by regulating the TLR4/MyD88/NF-κB/NLRP3 inflammasome pathway. Moreover, a study indicated that trimetazidine could regulate pyroptosis by targeting GSDMD and playing a vital role in suppressing the expression of TLR4, MyD88, phosphorylated NF-κB p65, and NLRP3 inflammasome [108].

### 6.3. Natural Substances

Natural substances primarily include traditional Chinese medicinal compounds, crude extracts of traditional Chinese medicines, and active monomer components. Generally, they have multiple components, multiple channels, and multi-target effects, and show fewer side effects on patients. Numerous studies have shown that natural substances offer unique advantages and disadvantages for ischemic cardiomyopathy and have complex pathological mechanisms and therapeutic potentials in cardiovascular diseases.

Emodin exhibits excellent anti-inflammatory effects, which can protect the myocardium from MIRI. It was observed that emodin inhibited the GSDMD-mediated TLR4/MyD88/NF-κB/NLRP3 inflammasome pathway for MIRI alleviation in cardiomyocytes [109]. Moreover, emodin led to an improvement in cell survival in vitro and a reduction in the size of myocardial infarction in vivo by inhibiting I/R-induced pyroptosis and decreasing the expressions of pyroptosis-related proteins.

Cinnamyl ethyl acetate extract has been shown to protect the heart of rats from MIRI by inhibiting NLRP3 inflammasome activation and pyroptosis [110]. The results from that study indicated that cinnamyl ethyl acetate extract reduced the size of myocardial infarction, improved myocardial function, and reduced the level of ASC, IL-1β, caspase-1, and GSDMD, and suppressed the expression of inflammatory factors in N-terminal GSDMD. Although the effects of natural substances for MIRI alleviation are well known, their clinically applications are rare due to the complex composition of natural substances, unknown risks, and few relevant clinical trials.

Myriocin is a specific inhibitor of serine palmitoyl transferase (SPT) and of ceramide de novo synthesis. During MIRI, SPT promotes the synthesis of ceramide, an inflammatory mediator, and makes it accumulate in myocardial cells to aggravate the damage. Myriocin can reduce inflammatory reaction and oxidative stress by downregulating ceramide and reducing the production of ROS [111,112]. Myriocin is effectively used as a post-treatment drug to reduce MIRI. Its role in immune and metabolic regulation is worthy of further exploration in the future.

### 6.4. Gases

Sevoflurane is one of the most commonly used inhalation anesthetics, which helps in rapid recovery and leads to a few adverse effects on circulatory disorders and respiratory depression in patients. Several studies have demonstrated that sevoflurane has a protective effect on MIRI by suppressing the inflammatory response. A review by Wu et al. indicated that the inhibition of cardiomyocyte inflammation and pyroptosis by regulation of the P2X7-NLRP3 signaling pathway is a potential mechanism of sevoflurane against MIRI [113]. Briefly, it was observed that sevoflurane treatment suppressed the high expressions of P2X7, NLRP3, ASC, caspase-1, and GSDMD and the release of LDH, CK, CK-MB, and MDA in the cells, and increased the activity of SOD. P2X 7 is a non-selective ion channel on the cell membrane, which opens when the extracellular ATP concentration is high, thus allowing for the passage of macromolecules and regulating the formation of the NLRP3 inflammasome. Therefore, during cardiomyocyte injury, there is an impairment of the mitochondrial function, and ATP released from the damaged or dying cells is recognized by P2X7, leading to the initiation of pyroptosis [114,115].

Hydrogen is an extremely flammable, colorless, transparent, odorless, and tasteless gas that is insoluble in water. Previous studies have indicated that hydrogen exerts antioxidant, anti-apoptotic, and anti-inflammatory effects in different types of diseases [116,117,118,119]. Similarly, hydrogen was shown to exert a certain protective effect on ischemia-reperfusion [120,121]. However, only a few studies focus on understanding the relationship between hydrogen and MIRI. In one of those studies, Nie reported that inhaled hydrogen led to a significant improvement in myocardial infarct size, no-return zone, cardiac function, microstructure, and mitochondrial morphology in a rat model of MIRI [122]. It was shown that the expression of pyroptosis-related proteins regulated by ROS and NLRP3 significantly decreased. Therefore, it can be concluded that hydrogen leads to MIRI alleviation by suppressing oxidative stress and NLRP3-mediated pyroptosis. Therefore, there is no significant improvement in the reflux caused by ischemia-reperfusion, which can have great potential benefits for the treatment of patients with ischemic cardiomyopathy. More studies must be conducted to understand the application and mechanism of action of hydrogen in MIRI.

The protective effect of small molecular substances on MIRI is governed by their chemical properties. Generally, these substances have dose-dependent or time-dependent characteristics and show only a few adverse effects on humans or rats. NLRP3 inhibitors may also be indirectly involved in MIRI regulation because of their action on other pathways. This can provide new insights into the treatment and prevention of ischemic cardiovascular diseases. The molecular inhibitors involved in other signaling pathways may also contribute to the inhibition of inflammatory responses and cell pyroptosis, thus further advocating the need to evaluate the efficacy and mechanism of action of acute myocardial infarction (Table 2). Therefore, at present, the combination of clinical drugs and small molecular substances for better curative effects has become a research hotspot. However, the timing, duration, targets, and interaction effects of these drugs remain uncertain and need to be studied in detail.

## 7. Thinking and Improvement

Myocardial ischemia-reperfusion is a complex pathophysiological process involving multiple molecular signaling pathways. A complete understanding of MIRI development and progression can help to identify targets for the treatment of patients with associated cardiovascular diseases and improve the clinical prognosis of patients with acute myocardial infarction and coronary heart disease after PCI. Several mechanisms of MIRI are closely linked and dependent on each other. Some of the mechanisms, such as calcium overload, energy metabolism disorders, oxidative stress, autophagy, etc., have been largely studied and are demonstrated to exhibit a regulatory role in MIRI development. The role of pyroptosis in MIRI is still under investigation, and the associated signaling pathways need to be elucidated further. Recently, a majority of studies on MIRI alleviation have focused on improving pyroptosis by inhibiting NLRP3 inflammasome, and NLRP3 is proven to be a potentially effective key molecule in MIRI treatment. A cascade of inflammatory responses in MIRI is triggered by the release of several inflammatory factors, leading to myocardial infarction and myocardial fibrosis. However, the inflammatory response is indispensable to promoting repair sites of damage. A proper balance of inflammatory and anti-inflammatory responses can help in the timely removal of necrotic tissue cells from the human body. The studies on the regulatory role of pyroptosis in MIRI alleviation revealed that the degree of pyroptosis can be controlled for the regulation of the inflammatory responses in tissues and repairing the damaged myocardium to obtain an optimal degree for MIRI alleviation. It is noteworthy that cardiac fibroblasts act as sentinel cells, and NLRP3 inflammasome is upregulated in cardiac fibroblasts to mediate MIRI, whereas cardiomyocytes are mere “victims” of pyroptosis. However, further experimental studies are required to support and verify this conclusion.

Although several inhibitors of pyroptosis have been used in MIRI animal models and have achieved good efficacy in reducing myocardial infarction size and improving cardiac functions, the present studies are still limited to animal experimental models, and the human clinical trials to understand the effect of these inhibitors remain insufficient. At present, all of these drugs are not approved for use in MIRI patients. Therefore, it is expected that more clinical trials to verify the results can be helpful to promote the research on potential drugs. This study also elucidated the effects of different drug categories, including small molecule drugs, clinical drugs, proprietary Chinese medicines, and even gases, on MIRI and cell pyroptosis. They act in different overlapping signaling pathways and ultimately affect cells. Therefore, the combined effect of two different drugs on MIRI treatment needs to be further studied to promote their applications in clinical trials. Perhaps, with the advancements in the pharmacodynamics, pharmacokinetics, toxicology, and corresponding mechanisms of drugs, studies can be conducted to understand the applications of binding the useful and effective monomer components of different drugs, which may provide a breakthrough innovation in MIRI treatment. Pyroptosis is involved in the occurrence and development of several cardiovascular and inflammatory-related diseases. Further research on the pyroptosis signaling pathway of diabetics with MIRI can help us clearly understand the specific mechanism of pyroptosis in such disease and provide a new direction for MIRI prevention and treatment.

## 8. Summary and Outlook

Myocardial ischemia-reperfusion injury is a deleterious process involving multiple signaling pathways, and affecting multiple genes, multiple molecules, multiple cells, and multiple tissues. The underlying mechanisms are driven by biological signal transduction pathways and epigenetics. Pyroptosis interacts with other models of cell death in MIRI, and out of these, the potential therapeutic effects of inhibiting the major targets of NLRP, caspase, and GSDMD have gradually been elucidated. However, at present, no available drug can inhibit the expression of proteins associated with pyroptosis for the clinical treatment of myocardial ischemia-reperfusion syndrome. Furthermore, sevoflurane and hydrogen have great potential applications in MIRI treatment. Targeting pyroptosis helps to strictly control the degree of inflammatory response; however, a few issues still need to be resolved to realize such applications. Due to the complex and diverse characteristics of MIRI pathogenesis, single-target therapy for reperfusion injury may be less effective. In such cases, multi-target combination therapy may be an effective approach for the treatment of reperfusion injury. Therefore, it is necessary and urgent to discover promising therapeutic targets for maximizing the benefits of revascularization.

## Figures and Tables

**Figure 1 cells-11-03271-f001:**
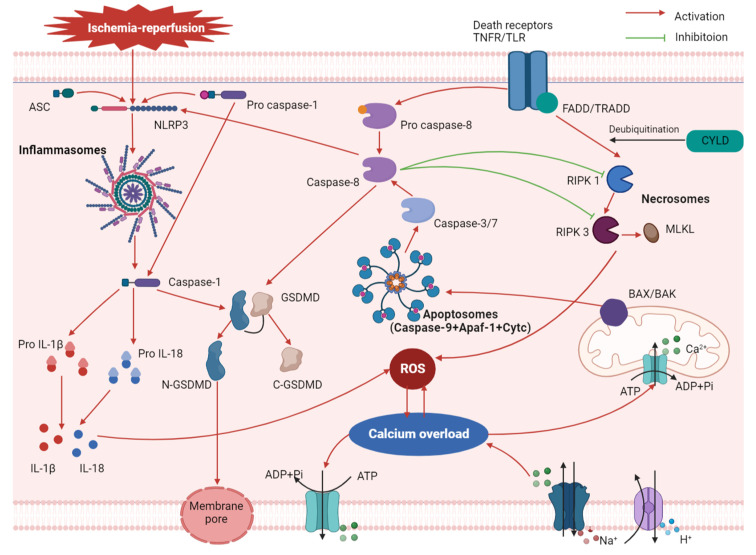
The cross-link between pyroptosis and other mechanisms (oxidative stress, calcium overload, apoptosis, and necroptosis) in myocardial ischemia-reperfusion injury.

**Table 1 cells-11-03271-t001:** Non-coding RNAs affect myocardial ischemia-reperfusion injury by regulating pyroptosis.

Non-Coding RNAs	Experimental Model	Targeted Gene	Expression with MIRI	Mechanism	First Author/Year
MiRNA-29a	Myocardial cells hypoxia/ reoxygenation model	SIRT1	Increased	Activate oxidative stress and NLRP3-mediated pyroptosis pathway	Ding et al. 2020 [61]
MiRNA-29b	MIR rat and myocardial cells hypoxia/ reoxygenation model	FoxO3a/ARC	Increased	Activate pyroptosis and inflammatory reaction	Zhong et al. 2020 [73]
MiRNA-132	MIR rat and myocardial cells hypoxia/ reoxygenation model	SIRT1	Increased	Activate oxidative stress and pyroptosis	Zhou et al. 2020 [62]
MiRNA-383	MIR rat model	RP105	Increased	Activate cardiomyocyte pyroptosis	Guo et al. 2021 [74]
LncRNA Rian	MIR rat and myocardial cells oxygen-glucose deprivation/reoxygenation model	MiRNA-17-5p	Decreased	Reduce cardiomyocyte pyroptosis	Kang et al. 2022 [65]
LncRNA ROR	Myocardial cells hypoxia/ reoxygenation model	MiRNA-185-5p	Increased	Activate cardiomyocyte pyroptosis	Sun et al. 2022 [66]
LncRNA PVT1	MIR rat and myocardial cells hypoxia/ reoxygenation model	GSDMD	Increased	Activate cardiomyocyte pyrotosis	Li et al. 2021 [67]
CircRNA-NNT	MIR rat and myocardial cells hypoxia/reoxygenation model	MiRNA-33a-5p	Increased	Activate cardiomyocyte pyroptosis	Ye et al. 2021 [72]

**Table 2 cells-11-03271-t002:** Drugs alleviate myocardial ischemia-reperfusion injury by affecting pyroptosis.

Drug	Experimental Model	Mechanism of Action	Effect	First Author/Year
MCC950	In a vivo pig model of myocardial infarction	Selectively inhibits NLRP3-inflammasome formation and reduces pyroptosis, IL-18, and IL-1b signaling.	Reduces infarct size circulating markers of damage and inflammation and the influx of myocardial neutrophil and preserves cardiac function.	Hout et al. 2017 [77]
INF4E	In a vitro rat model of MIRI	Inhibits the NLRP3 inflammasome, activates the prosurvival RISK pathway, and improves mitochondrial function.	Reduces infarct size and lactate dehydrogenase release and improves the ventricular pressure in the postischemic left.	Mastrocola et al. 2016 [78]
VX-756	In a vitro rat model of MIRI	Selectively inhibits prodrug caspase 1 and actives the PI3K/Akt pathway (the reperfusion injury salvage kinase (RISK) pathway).	Reduces infarct size.	Carmo et al. 2018 [82]
Colchicine	In a vivo rat model of MIRI	Increases the level of IL-10 and decreases the level of cardiac TGF-β.	Reduces infarct size and inhibits the increased expression of inflammatory cytokines.	Bakhta et al. 2018 [87]
Metformin	In a vitro rat model of MIRI and ventricle myocytes hypoxi/reoxygenation model	Enhances the AMPK pathway and suppresses the activation of NLRP3 inflammasome.	Alleviates myocardial infarct size, attenuates cell apoptosis, inhibits myocardial fibrosis, and decreases the level of pro-inflammatory cytokines, such as TNF-α, IL-6, and IL-1β, as well as decreases the activation of NLRP3 inflammasome.	Zhang et al. 2020 [103]
Dexmedetomidine	In a vivo rat model of MIRI and ventricle myocytes hypoxi/reoxygenation model	Downregulating miR-29b to activate FoxO3a/ARC axis to attenuate cell pyroptosis and ameliorate inflammatory response.	Reduces the size of myocardial infarction and the expression levels of cellular inflammation and pyroptosis-related proteins or markers.	Zhong et al. 2020 [73]
Trimetazidine	In a vivo rat model of MIRI and ventricle myocytes hypoxi/reoxygenation model	Alleviates pyroptosis through the TLR4/MyD88/ NF-κB/NLRP3 inflammasome pathway.	Increases the viability of cardiomyocytes, reduces the infarct size, and inhibits noncanonical inflammasome signaling.	Chen et al. 2022 [108]
Emodin	In a vivo rat model of MIRI and ventricle myocytes hypoxi/reoxygenation model	Alleviates pyroptosis through the TLR4/MyD88/ NF-κB/NLRP3 inflammasome pathway.	Increases the rate of cell survival in vitro and decreases the myocardial infarct size in vivo.	Ye et al. 2019 [109]
Cinnamyl ethyl acetate	In a vivo rat model of MIRI	Suppresses NLRP3 inflammasome and subsequent pyroptosis-related signaling pathways.	Decreases myocardial infarct size and improves cardiac function, mitigates myocardial damage, and represses inflammatory response.	Peng et al. 2021 [110]
Sevoflurane	Patients with a history of myocardial ischemia who underwent abdominal surgery with Sevoflurane general anesthesia and ventricle myocytes hypoxi/reoxygenation model.	Inhibits the expression of IL-1β, IL-18, and GSDMD by inhibiting the P2X7-NLRP3 signaling pathway to regulate inflammatory reaction and pyroptosis.	Reduces myocardial infarct size, the expression of inflammatory factors, and infiltration of inflammatory cells.	Wu et al. 2022 [113]
Hydrogen	In a vivo rat model of MIRI	Inhibits oxidative stress and NLRP3-mediated pyroptosis.	Improves myocardial infarct size, no-reflow area, cardiac function, microstructure, and mitochondrial morphology.	Nie et al. 2021 [122]

## Data Availability

Not applicable.

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
