# Peer review of "Research Progress on the Role of Pyroptosis in Myocardial Ischemia-Reperfusion Injury"

_cells, 2022, doi:10.3390/cells11203271_

Round 1
Reviewer 1 Report
The Review entitled "Research progress and prospects of pyroptosis in myocardial ischemia-reperfusion injury" by Yang Liu et al. is a comprehensive analysis of the data published in recent years on the molecular mechanisms of pyroptosis in cardiovascular pathologies. It is an interesting collection of information organised into large, important chapters. The topic is potentially very promising. However, the manuscript cannot be published as is. The authors list the data from the literature without giving particular detailed information and perspectives, and the reader is dealing with a collection of information that is not guided by a clear hypothesis. Therefore, it is quite difficult to follow the main text and focus on the most important messages. Although the amount of information is reliable and interesting, the authors should make an effort to reduce the data presented (I suggest deleting the chapters on noncoding RNAs and diabetes) to better and more thoroughly describe the molecular mechanisms presented. As it stands, the manuscript sounds like a simple summary of the literature. Finally, the entire manuscript needs extensive English proofreading. I suggest carefully read the main text to remove typos and improve grammar and syntax.
Minor points:
- the abstract should be extensively revised.
- The first sentence of the introduction is very long. Divide it and add some precise citations.
- Line 57: "U" remove uppercase
- Line 74: Modify the sentence adding comas: This, in turn, activates...
- Please put the full stop after the brackets for citations.
- The quality of the figure is very low. Could the authors try to make it more appealing?
Author Response
Thanks a lot for the reviewer detailed review and pointing out the problems from the whole perspective of the manuscript. We appreciate reviewer's agreement and doubts about our work. We did a lot of literature data collection work on this review, hoping to let people understand the current relationship and progress between pyroptosis and MIRI from various aspects. But people may not be able to extract all the key information. Therefore, all our authors have revised the manuscript through joint discussion, appropriately deleted some duplicate experimental examples, and added many sentences of views and summaries. "Thinking and Improvement" and "Summary and outlook" sections are conducive to the understanding of the manuscript. We hope reviewer are satisfied with the content. In addition, we deleted the content about pyroptosis and diabetes. We agree with the reviewers' view and think that the content is not closely related to the theme of the review. However, we still choose to retain the content of non-coding RNAs and pyroptosis. There is a lot of evidence that non-coding RNA regulates pyroptosis in MIRI through axial feedback regulation, and non-cardiomyocyte derived exosomes can carry microRNAs to inhibit pyroptosis pathway. Therefore, we think that non-coding RNAs is closely related to pyrotosis and inspire people to further explore more non-coding RNA to improve MIRI by mediating pyroptosis signal pathway. We also refined and summarized this part, hoping to get reviewer's understanding. We are very sorry that our weak English has caused reading obstacles and confusions about the content. This is where we need to study hard all the time. Our revised manuscript has been edited by the professional English polishing company, but we are also aware of our shortcomings in English writing. Thanks again for reviewer's patience and careful review, helping us improve the content of the manuscript. We hope the revised manuscript can be satisfied by reviewers.
Minor points:
- the abstract should be extensively revised.
Thank you for your suggestion. We have completely revised the abstract in terms of content and language, and we hope it can meet your requirements.
- The first sentence of the introduction is very long. Divide it and add some precise citations.
We have divided the acute myocardial infarction definition in the first sentence of the introduction into two sentences and inserted two relevant references.
Revised sentence (line 35-38): Acute myocardial infarction (AMI) is a severe and common medical emergency, resulting in irreversible damage to the heart (1). AMI leads to ischemia and necrosis of the corresponding myocardial area due to acute and persistent stenosis of the underlying coronary arteries (2).
The references are as follow: (1) Anderson JL, Morrow DA. Acute Myocardial Infarction. The New England journal of medicine. 2017;376(21):2053-64. (2)Algoet M, Janssens S, Himmelreich U, Gsell W, Pusovnik M, Van den Eynde J, et al. Myocardial ischemia-reperfusion injury and the influence of inflammation. Trends in cardiovascular medicine. 2022.
- Line 57: "U" remove uppercase
We are so sorry for such a simple mistake. We have corrected it. We've replaced it with another word “elucidating” and think it's more appropriate. Thanks again for your careful review.
- Line 74: Modify the sentence adding commas: This, in turn, activates...
Thank you for your suggestion. We have added commas to make sentences conform to specifications.
- Please put the full stop after the brackets for citations.
We are very sorry for the mistake and we checked the full text again and have added full stops to all references in the manuscript. Thank you again for your careful review, which is a great help to improve our manuscript.
- The quality of the figure is very low. Could the authors try to make it more appealing?
We have redrawn the mechanism figure. We hope you think it meets the requirements of the journal. Thanks for your comments.

Reviewer 2 Report
This review article discusses pyroptosis' role during myocardial ischemia-reperfusion injury (MIRI). This is a complicated topic with so many interplays of pyroptosis with other mechanisms of MIRI. The authors have done a great job of summarizing the evidence. The Conclusion or perspective and future direction section may be improved to combine the "Thinking and Improvement" and "Summary and outlook" sections.
Minor:
Line 189: "Regarding other mechanisms of MIRI, such as autophagy, endoplasmic reticulum 189 stress, and ferroptosis, there are few studies available and will not be stated here. "There are a lot of studies on MIRI involving autophagy, ER stress, and ferroptosis. I think what you mean is that there are few studies connecting these processes with pyroptosis. Please clarify.
Line 211: Are these NLRP3 knockout mice cardiomyocyte or fibroblast specific? Please clarify.
Line 239: These are the role of pyroptosis in the coronary artery disease prevention. Please remove or state it clearly in a separate paragraph.
Table 1. LncRNA PVT1 and CircRNA-NNT, please specify what are increased in the column of Mechanism.
Line 489: What is a no-return zone? Please define.
Author Response
Thanks very much for reviewer's approval of our work, which is a great encouragement for all our authors. We have revised and enriched the summary of current work and the future direction about pyroptosis in the manuscript again. We hope reviewers can be satisfied with our revision.
Minor:
Line 189: "Regarding other mechanisms of MIRI, such as autophagy, endoplasmic reticulum 189 stress, and ferroptosis, there are few studies available and will not be stated here. "There are a lot of studies on MIRI involving autophagy, ER stress, and ferroptosis. I think what you mean is that there are few studies connecting these processes with pyroptosis. Please clarify.
Thanks for your understanding. What we want to express is that few studies on these mechanisms are associated with pyrotosis in MIRI. We have revised this sentence in the manuscript to express our views clearly.
Revised sentence: There are few studies on the relationship between pyroptosis and other mechanisms of MIRI, such as autophagy, endoplasmic reticulum stress, and ferroptosis. So, those studies are not considered here. (line 193-195)
Line 211: Are these NLRP3 knockout mice cardiomyocyte or fibroblast specific? Please clarify.
Thank you for your careful review and allowing us to clarify this issue. These mice were able to knock out the NLRP3 gene in fibroblasts,so NLRP3 gene was not expressed in fibroblasts. Because in the "Discussion" section of the cited literature 47 (The NLRP3 inflammasome is up-regulated in cardiac fibroblasts and mediates myocardial ischaemia-reperfusion injury.), there was written "but in this model IL-1 β release will solely depend on myocardial cells.”. The expression level of NLRP3 gene was the highest in fibroblasts, but after knockout, the pyrotosis pathway could not be activated, so IL-1 β was only released by myocardial cells.
Revised sentence: After NLRP3 was knocked down in fibroblasts of rats, the contractile function of the hearts was preserved after in vitro ischemia-reperfusion, and the infarct size and apoptosis were significantly reduced compared to the control. (line 213-216)
Line 239: These are the role of pyroptosis in the coronary artery disease prevention. Please remove or state it clearly in a separate paragraph.
Thanks for your suggestions. We agree with you and have deleted these sentences about the role of pyrotosis in the prevention of coronary artery disease.
Table 1. LncRNA PVT1 and CircRNA-NNT, please specify what are increased in the column of Mechanism.
We are sorry for the simple error. We have revised the mechanism of LncRNA PVT1 and CircRNA NNT to "Activate cardiomyocyte pyroptosis".
Line 489: What is a no-return zone? Please define.
We are very sorry for the confusion caused by our incorrect English expression. "No return zone" is an inappropriate expression. We have revised it to the formal expression "no-reflow area". No-reflow area refers to the area where ischemia continues without adequate perfusion. This is a common phenomenon in ischemia-reperfusion named "no reflow". The continuity and superposition of ischemic area lead to the aggravation of myocardial injury.

Round 2
Reviewer 1 Report
Dear authors,
I greatly appreciate your efforts to improve your manuscript. In its present form it sounds more accurate and clear. The English has been significantly improved and reading is much easier and smoother.
I have only one suggestion for the section on natural substances. It has been shown that myriocin, a non-proteinogenic amino acid derived from certain fungi, also has a protective effect against MIRi by reducing cardiomyocyte death and inhibiting inflammation. It is worth mentioning this molecule. Here are some references:
- Bonezzi F, Piccoli M, Dei Cas M, Paroni R, Mingione A, Monasky MM, Caretti A, Riganti C, Ghidoni R, Pappone C, Anastasia L, Signorelli P. Sphingolipid Synthesis Inhibition by Myriocin Administration Enhances Lipid Consumption and Ameliorates Lipid Response to Myocardial Ischemia Reperfusion Injury. Front Physiol. 2019 Aug 9;10:986. doi: 10.3389/fphys.2019.00986. PMID: 31447688; PMCID: PMC6696899.
- Reforgiato MR, Milano G, Fabriàs G, Casas J, Gasco P, Paroni R, Samaja M, Ghidoni R, Caretti A, Signorelli P. Inhibition of ceramide de novo synthesis as a postischemic strategy to reduce myocardial reperfusion injury. Basic Res Cardiol. 2016 Mar;111(2):12. doi: 10.1007/s00395-016-0533-x. Epub 2016 Jan 19. PMID: 26786259. Overall, the manuscript can be accepted after minor revisions.Author Response
Reviewer 1:
Thank you very much for your affirmation of our revised manuscript. All our authors are very happy and inspired. Again, thank you for all your suggestions on our manuscript. We gradually improve our manuscript because of your careful review and we will continue to improve our deficiencies in the future.
Thank you very much for your suggestions. We have added relevant content of myiocin to the section of natural substances (line 586-592). This is an interesting and meaningful finding. Myiocin can alleviate MIRI by inhibiting inflammatory reaction, which is also related to immunity and metabolism. We all agreed to add it to our manuscript. At present, there is no literature report linking myiocin with pyroptosis clearly. Although it has the effect of inhibiting inflammation, the mechanism of pyroptosis with myiocin need to be found in the future. This is still a topic to explore. Thank you again for your supplement and help.
